# A *Lipoxygenase 3* mutation reverses growth phenotypes in an Arabidopsis *Plastid Lipase 3* overexpression line

Yosia Mugume[1], Ron Cook[1], Breana Hagerty[1], Jinjie Liu[1,2], Zachary B. Alvord[1], Linda Danhof[1], John E. Froehlich[1,3], Josh V. Vermaas[1,3], Christoph Benning[1,3,4]*

**1** MSU-DOE Plant Research Laboratory, Michigan State University, East Lansing, Michigan, United States of America, **2** Biological Sciences Program, Michigan State University, East Lansing, United States of America, **3** Department of Biochemistry and Molecular Biology, Michigan State University, East Lansing, Michigan, United States of America, **4** Department of Plant Biology, Michigan State University, East Lansing, Michigan, United States of America,

* benning@msu.edu

## Abstract

Plastid Lipase 3 (PLIP3) is a chloroplast phospholipase A that cleaves linolenic acid, a polyunsaturated fatty acid, from the *sn*-1 position of the chloroplast membrane lipid monogalactosyldiacylglycerol. Linolenic acid is subsequently converted by enzymes in the chloroplast and the peroxisome to jasmonic acid (JA) requiring transport between the organelles. Overexpression of a cDNA encoding PLIP3 resulted in stunted plant growth with altered leaf morphology caused by accumulation of JA and other oxylipin metabolites redirecting the metabolism from growth to defense. We conducted a genetic suppressor screen in the *PLIP3*-OX (*PLIP3* overexpression) line to query the entire pathway from the start of JA biosynthesis to its perception, transduction, and modification by other signaling pathways. We identified a mutant allele of the 13-lipoxgenase LOX3, *lox3–4*, that is causal to the attenuation of the *PLIP3*-OX phenotype with reduced dwarfism and decreased production of JA and other oxylipins. The responsible G776E point mutation is in the C terminal catalytic domain in proximity to the non heme iron binding site of LOX3. The point mutation likely inhibits the oxidation of α-linolenic acid demonstrating its importance for general JA biosynthesis as its activity cannot be compensated for by other 13-lipoxygenases.

## Introduction

Chloroplasts are cellular organelles that are the site for photosynthesis in plants. Their membranes are composed of the phospholipid phosphatidylglycerol (PG), the sulfolipid sulfoquinovosyldiacylglycerol (SQDG), and the galactolipids monogalac-tosyldiacylglycerol (MGDG) and digalactosyldiacylglycerol (DGDG), of which the latter three are found primarily in photosynthetic organisms [1,2]. Chloroplasts harbor the machinery for photosynthesis and other biochemical pathways, e.g., fatty acid

**Data availability statement:** All relevant data are within the paper and its Supporting Information files.

**Funding:** This work was primarily supported by the National Science Foundation (MCB-2203474) and in part by the Office of Basic Energy Sciences of the United States Department of Energy (Grant DE-FG02-91ER20021) and by MSU AgBioResearch (MICL02632). Z.B.A. was supported by the High School Honors Science, Math and Engineering Program at Michigan State University. This work was supported in part through computational resources and services provided by the Institute for Cyber-Enabled Research at Michigan State University. The funders had no role in study design, data collection and analysis, decision to publish, or preparation of the manuscript.

**Competing interests:** The authors have declared that no competing interests exist.

(FA) biosynthesis, and respond to stress by lipid remodeling [3–5], and synthesis of defense signaling compounds such as jasmonic acid (JA) [6].

In response to environmental and developmental signals, chloroplast membrane glycerolipids are broken down by plastid lipases to release acyl chains or the head groups from the glycerol backbone [7–9]. Lipases are classified based on their preferred hydrolysis position, for example Phospholipase A (PLA) releases FAs from the glyceryl sn-1 or sn-2 positions of phospholipids; PLA1 hydrolyzes at the sn-1 position, while PLA2 hydrolyzes at sn-2 [10]. Plastid Lipase 3 (PLIP3) is a chloroplast-resident glycerolipid A1 lipase that releases α-linolenic acid, an 18:3 acyl group (carbons: double bonds), from chloroplast membrane lipids [8]. Overexpression of the *PLIP3* cDNA in the Arabidopsis (*Arabidopsis thaliana*) *PLIP3*-OX transgenic line leads to constitutive oxylipin accumulation including jasmonic acid (JA), an important signaling molecule produced from 18:3 acyl groups. Consequently, these *PLIP3*-OX transgenic plants show stunted growth and upregulation of JA-responsive genes which can be reversed by genetically blocking JA perception through the introduction of the *coi1* JA receptor mutation, thus confirming the role of PLIP3 in JA production [8].

FAs released by plastid lipases are oxygenated by lipoxygenases (LOXs) [11] to yield FA hydroperoxides collectively known as oxylipins. Oxylipins are routed through different metabolism branches as precursors for functionally diverse compounds in response to various developmental and environmental cues, e.g., [12]. JA is one such oxylipin synthesized by a pathway involving allene oxide synthase (AOS) and allene oxide cyclase (AOC) to form 12-oxo phytodienoic acid (OPDA). OPDA is exported from the plastid involving the transport protein JASSY [13] and then imported into the peroxisome by an acyl-CoA transporter [14–16] for further processing before JA is routed back to the cytosol for conjugation with the amino acid isoleucine to form the bioactive JA-Ile [6], which is imported into the nucleus [17]. This conversion of chloroplast FAs to JA likely occurs constitutively in *PLIP3*-OX, due to the PLIP3-mediated release of 18:3 FA precursors from membrane lipids, resulting in its distinctive JA-induced phenotype [8].

Lipoxygenases are non-heme iron or manganese-containing dioxygenases and occur in most eukaryotes and in some prokaryotes [18]; they are made up of two distinct domains namely N-terminal polycystin-1 - lipoxygenase - alpha-toxin - triacylglycerol lipase (PLAT) domain, whose function is not yet fully understood, and the C-terminal catalytic LOX domain. These enzymes are classified as 9- or 13-lipoxgenases in accordance with the position of oxygen incorporation in an 18-carbon acyl group [19]. Moreover, lipoxygenases with dual 9-and 13-lipoxygenase activity have also been reported [20]. Six lipoxygenases are encoded by the Arabidopsis genome; LOX1 and LOX5 are 9S-lipoxygenases oxygenating both linoleic and linolenic acid with similar activity, and LOX2, LOX3, LOX4 and LOX6 are 13S-lipoxygenases preferentially oxygenating 18:3 FAs [19], although AtLOX2 also has significant activity towards 18:2 acyl groups [18]. While all 13S-LOXs potentially contribute to JA production, different genes encoding LOX-orthologs are expressed in a stress-dependent manner. For example, *At*LOX1 and *At*LOX5 are linked to lateral root development and defense responses [19]

while the *AtLOX3* and *AtLOX4* genes respond to wounding and are important for reproductive development [21,22]. *At*LOX2 is essential for JA biosynthesis in response to wounding while *At*LOX6 seems more related to JA pathogen defense pathways and is generally present in roots [19]. Thus, different 13S-LOX isoforms have distinct roles in vegetative and reproductive development and stress responses.

Although many aspects of JA metabolism and signaling are known, the role of different oxylipins and the integration of different hormone responses with JA signaling remains to be further explored. To query the entire pathway from the release of the fatty acid precursor and the biosynthesis of JA and other oxylipins to their subsequent perception, transduction, and modification by other signaling pathways, we performed a suppressor screen in the *PLIP3*-OX transgenic line as part of a course based undergraduate research experience [23,24]. If any aspect of oxylipin biosynthesis, perception, or signal transduction is disrupted in the *PLIP3*-OX line by a secondary mutation, we would expect the phenotype of the suppressor line to revert to the wild-type phenotype. To aide in the genetic analysis, it was necessary to identify the insertion location of the *PLIP3* transgene in the *PLIP3*-OX transgenic line. The transgenic line contains elevated oxylipins and JA levels with distinctive JA-induced phenotypes such as stunted growth, altered relative dimensions of leaves and petioles, and anthocyanin accumulation in vascular tissues [8]. Mutagenized *PLIP3*-OX plants were screened for suppression of the JA-induced phenotypes. We identified a suppressor mutant Sup52, which harbors a glycine-to-glutamic acid mutation at position 776 of the polypeptide chain of LOX3, that we designated *lox3–4*. This mutant provides the first reported phenotype for a single *lox3* mutant allele recovered in a forward genetics screen [25].

## Materials and methods

### Plant materials and growth conditions

All experiments were performed in the *Arabidopsis thaliana* Col-0 ecotype as the wild-type (WT) control. T-DNA insertion mutants used in this study were SALK_062064 (*lox 3−3*) and SALK_119404 (*lox3−1*) [21] and were obtained from the Arabidopsis Biological Resource Center, Ohio State University and the *PLIP3*-OX was as previously described [8] and available in the Benning Lab stock. Plants were grown in SUREMIX™ Professional All-Purpose Perlite Mix (Michigan Grower Products, Inc.) at 22°C and a light intensity of approximately 90 μmol $m^{-2}$ $s^{-1}$, under a 16/8-hr light/dark cycle in a growth chamber [26].

### Point mutant generation

The procedure for the mutagenesis was done as previously reported [27]. Approximately 13,000 *PLIP3*-OX seeds were incubated in 0.1% Tween20© (Sigma-Aldrich) for 15 minutes in a tube rotator, after which seeds were allowed to settle, and the solution was removed. The seeds were thereafter treated with 25 volumes of 0.2% (v/v) ethyl methane sulfonate (EMS, Sigma-Aldrich) in a 50-ml conical tube for ~16 h with rotation after which the EMS solution was discarded using a pipette. After seven washes with distilled water, seeds were soaked in distilled water for 2 hours to allow the EMS to diffuse out and washed one more time. M1 seeds were sown on soil and grown under the growth conditions described above. M2 seeds were collected in 96 pools of M1 plants.

### Bulked Segregant Analysis

To perform bulk segregant analysis, we created an F2 population by backcrossing Sup52 to *PLIP3*-OX. Genomic DNA from pools of ≥ 100 four-week-old F2 plants with or without the suppressor phenotype (small or big plants) was extracted using the Promega Wizard® Genomic DNA Purification Kit according to the manufacturer's instruction. The final purified genomic DNA was quantified using both Qubit fluorometer dsDNA BS (Thermo Fisher Scientific, Carlsbad, CA) and NanoDrop (Thermofisher Scientific). Equal amounts based on concentration of DNA from 50-200 plants were pooled based on phenotype and then sent to Innomics Inc (https://www.innomics.com/) for sequencing with their paired end DNBseq™

platform. At least 10 Gb were sequenced for each sample. Sequencing data was processed using the SIMPLE pipeline [28] to generate single nucleotide polymorphisms (SNPs) and identify candidate genes.

## T-DNA insertion site determination using TC-hunter

We used TC-hunter [29] to locate the site of the T-DNA insertion in *PLIP3*-OX. The software was obtained from GitHub (https://github.com/vborjesson/TC_hunter), though we applied some fixes to the pipeline file so that it will run with current Nextflow [30] versions. In brief, TC-hunter uses Burrows-Wheeler transform [31] to map reads against genomic and insert sequences, then extracts reads which map across both genomic and insert sequences (chimeric) as well as read pairs where one read maps to the genome and the other to the insert (discordant). This process was done independently for two different mapping population sequence data sets from plants homozygous for the *PLIP3*-OX insertion locus available in the lab. For the first, we identified 14 chimeric reads near the left end of the insertion, with 13 mapping to the same breakpoint (54108 bp on chromosome 5 (Gene Bank ID CP002688.1) to 444 bp on the insert), with 29 supporting discordant pairs. For the second sequencing data set, we found 11 chimeric reads which also mapped to the same breakpoint at 444 bp on the insert, with 52 supporting discordant pairs. The second data set had two additional chimeric reads, but these mapped to different chromosomes and insert position with < 10 supporting discordant pairs and were not considered.

## Quantitative PCR (qPCR)

Total RNA was extracted from 3-weeks-old plants using the RNeasy Plant Mini Kit (Qiagen, Cat. No. 74904) according to the manufacturer's instructions. cDNA was synthesized from 1000 ng of RNA using the iScript™ cDNA Synthesis Kit (Bio-rad). Fast SYBR Green Master Mix (Applied Biosystems) was used in the presence of gene-specific primers and template cDNAs in an ABI7500 (Applied Biosystems). The primer pairs used for UBQ10 (AT4G05320, used for normalization purposes), *VSP1* (At5g24780), *PDF1.2* (At5g44420), and *LOX2* (AT3G45140) are listed in S1 Table.

## Measurement of anthocyanins

The method for anthocyanin measurement was modified from [32,33]. About 10 mg of leaf tissue was collected from 1 month old plants in pre-labeled 2 mL cryo tubes (USA Scientific, Cat#1420–9600) containing about 3–5 3 mm zirconium beads (Glen Mills, Cat# VHD ZrO), and flash-frozen in liquid nitrogen. Samples are then placed in pre-frozen beat adaptors (TissueLyzer II, Qiagen) and completely ground at 30/s for 3 minutes. 1 mL of extraction buffer (45% methanol, 5% acetic acid v/v) was added and thoroughly mixed by vortexing. This was followed by centrifugation at 12,000 x g for 5 minutes. Absorbance was measured at 530 and 657 nm using a spectrophotometer. Results were normalized to WT=1.

## Hormone quantification

Fresh plant tissue ~500 mg was harvested in pre-labeled 2 mL cryo tubes (USA Scientific, Cat#1420–9600) containing about 3–5 3 mm zirconium beads (Glen Mills, Cat# VHD ZrO), their mass recorded and flash-frozen in liquid nitrogen. Samples were then placed in pre-frozen beat adaptors and completely ground at 30/s for 3 minutes. They were then incubated in extraction buffer (80:20 methanol: water, 0.1% formic acid, 100 mg/L butylated hydroxytoluene) containing internal standard (100 nM SA ($^{13}$C-6) JA (d-5) ABA (d-6) IAA ($^{13}$C6)) by rocking them for 24 hours at 4°C. Samples were analyzed using liquid chromatography/mass spectrometry and data analyzed as described by [8].

## Lipid profiling

The method for lipid profiling was modified from [34]. Tissue from 4-week-old soil-grown plants was harvested into a glass tube containing 1mL 1 M methanolic HCl and the tubes were sealed with teflon lined caps and incubated at 80°C for 40

minutes to derivatize them to fatty acid methyl esters (FAMEs). Afterwards, they were allowed to cool to room temperature, equal volumes of 0.9% (w/v) aqueous sodium chloride and hexane were added, and phases were separated after vortexing. The FAME-containing hexane phase was transferred to a new 1.5 mL falcon tube using a glass pipette, dried under $N_2$ gas, and resuspended with 100 µL hexane. FAMEs were identified and quantified using gas chromatography-flame ionization detection.

## Analysis of various LOX3 catalytic domain structures

The structure of the catalytic domain for Arabidopsis LOX3 (At1g17420; UniProt:Q9LNR3) residues 755–814; Camellia sativa LOX3 (Accession XP_010476891.1) residues 756–815; Zea mays (Zm) LOX3 (UniProt: Q8W0V2) residues 695–753; and Glycine max (Gm) LOX3 (UniProt: P09186) residues 694–753 were generated by AlphaFold3 [35]. The resulting catalytic domain structures for AtLOX3, CsLOX3, ZmLOX3 and GmLOX3 were then illustrated by ChimeraX [36,37] for structural comparison analysis.

## Molecular simulation methods

The starting structure for LOX3 was obtained via AlphaFill [38] based on the Uniprot [39] identifier (Q9LNR3) for the LOX3 structure from Arabidopsis, retaining the iron present in the predicted AlphaFill structure and predicted by Uniprot. The first 81 residues are part of the chloroplastic targeting sequence and are poorly predicted [40]. These 81 residues were thus cleaved when building the structure (S6A Fig). PROPKA 3.0 [41] was used to assign protonation states assuming pH 7.0 through pdb2pqr [40] prior to system construction in psfgen (https://www.ks.uiuc.edu/Research/vmd/plugins/psfgen/ug.pdf). Crucially, the iron is liganded to H578, H583, and H770, and the C-terminal carboxyl group on I919. This liganding process is analogous to what happens within photosystem II, and the parameters are thus taken by analogy from previous work [42]. From this step, the system construction bifurcates, with a G776E mutation added to facilitate comparisons to the wild type. The protein structures were solvated and ionized with 150 mM NaCl using the solvate and autoionize packages within VMD for a total system size of 210k atoms [43].

From this starting point, 5 independent simulation replicates were run for both the WT and LOX3 mutant protein simulation systems for 1 µs each, for an aggregate of 10 µs of simulation time. These simulations were carried out in NAMD 3.0b7 [44] using the CHARMM36m force field for proteins [45] together with modified parameters for the liganded iron [42]. Dynamics were calculated with a 12 Å cutoff, switched at 10 Å with a 1.2 Å grid spacing for the long-range electrostatics calculated by PME [46]. Each timestep was 2 fs, enabled by constraining covalent bonds to hydrogen via the SETTLE algorithm [47]. Langevin thermostats [48] and barostats were used to maintain 300K and 1 atm.

Analysis for the molecular simulation trajectories was performed in a python-enabled version of VMD 1.9.4b57, leveraging the numPy [49] and SciPy [50] libraries to generate plots in matplotlib [51]. This was used to quantify relevant quantities that can be tracked across the simulation set, such as the root mean square deviation (RMSD) to check for protein stability, as well as the root mean square fluctuation (RMSF) to evaluate especially dynamic parts of the protein.

## Accession numbers

GenBank (https://www.ncbi.nlm.nih.gov/genbank/about/) accession numbers for the proteins discussed are as follows:
*At*LOX3 - OAP14869.1; *Zm*LOX3 - NP_001105515.1; *Gm*LOX3 - NP_001235383.2, *Cs*LOX3 - XP_010476891.1; *At*LOX2 - OAP05844.1; *At*LOX4 - OAP12557.1
AtLOX6 - OAP12527.1
Arabidopsis DNA sequences can be found at TAIR (https://www.arabidopsis.org/):
   *HAKAI* - AT5G01160; *LOX2* - AT3G45140; *LOX3* – AT1G17420; *PDF1.2* AT5g44420; *UBQ10* - AT4G05320; *VSP1* - AT5g24780.

   

The GenBank accession number for the Arabidopsis chromosome 5 DNA sequence is CP002688.1.

## Results and discussion

### Suppressor mutant screen design and insertion location of the PLIP3-OX transgene

Three different PLIP overexpression lines, *PLIP1*-OX, *PLIP2*-OX and *PLIP3*-OX have been previously described [8,52]. All of these present a JA overproduction phenotype with *PLIP3*-OX having an intermediate phenotype, discernible from the wild type (WT) and maintaining fertility and seed production capacity (Fig 1A, S1 Fig). To identify new components related to chloroplast lipids and the JA pathway, we conducted a suppressor screen in a mutagenized M2 population of *PLIP3*-OX. We visually examined M2 plants grown on soil in 4-inch pots, five per pot, for approximately 3 weeks searching for a reversal of the *PLIP3*-OX phenotype. During the initial screen, 90 out of 5000 M2 plants showed increased size. We examined these for quantifiable phenotypes such as rosette diameter, ratio of leaf to petiole length, and ratio of leaf length to leaf width, alongside more subjective phenotypes such as anthocyanin content and distribution relative to *PLIP3*-OX.

To aid the subsequent mapping and phenotyping, we determined the insertion location of the transgene in *PLIP3*-OX using Transgene-Construct (TC) hunter [29]. Aside from determining whether the transgene insertion could cause interfering phenotypes by disrupting an endogenous gene, knowing the transgene insertion site allowed us to determine the zygosity at the *PLIP3*-OX insertion site and, thus, consider gene dosage effects during phenotypic analyses. For this

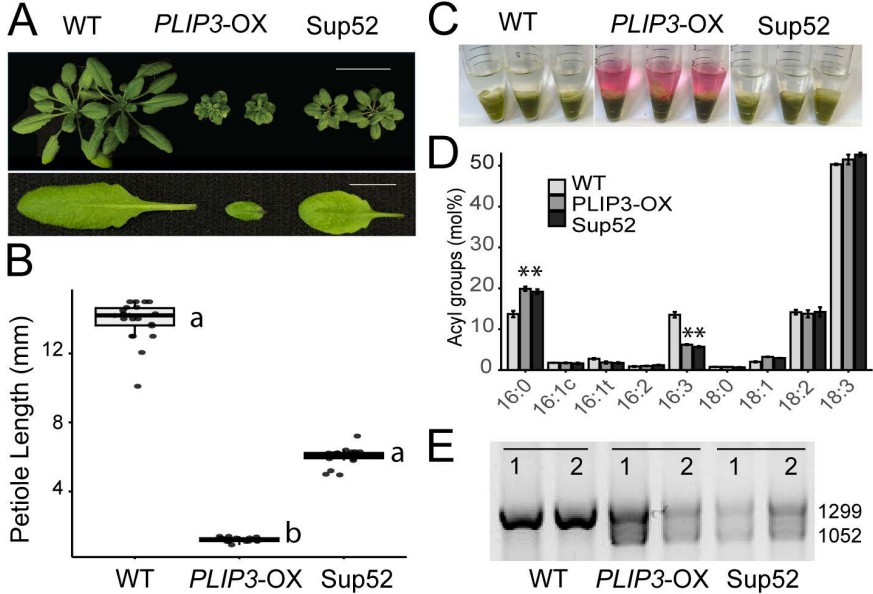

**Fig 1. Sup52 reverses the *PLIP3*-OX growth phenotype while maintaining activity of PLIP3. (A)** Image of 4-weeks-old plants from above showing the growth phenotype of wild-type (WT), *PLIP3*-OX and Sup52 homozygous lines. The scale bars represent 4 cm (plants) or 4 mm (leaves), respectively**.** Also shown is the leaf phenotype and representative petiole length from 4-week-old WT, *PLIP3*-OX and Sup52 plants. **(B)** Box plot showing petiole length of WT, *PLIP3*-OX and Sup52. n = 20. Statistical analysis was performed in R using ANOVA followed by Tukey's multiple comparison test to compare petiole length in WT to that of *PLIP3*-OX and Sup52 plants. The different letters indicate a difference of the means with p < 0.05. **(C)** Anthocyanin accumulation in *PLIP3*-OX following phase separation of the lipid extracts from plants grown under standard physiological conditions. Three independent extracts are shown for each genotype with the upper aqueous phases accumulating anthocyanin which manifests as red color. **(D)** Total acyl FA composition (mol%) of WT, *PLIP3*-OX and Sup52. n = 3, Student's t test was applied to compare WT plants with *PLIP3*-OX or Sup52 plants, respectively (*P < 0.05), which have the same genetic background responsible for the lipid phenotype; error bars show SD. **(E)** Gel image showing PCR genotyping results using *PLIP3* P1/2 primers flanking an intron in the *PLIP3* gDNA sequence. *PLIP3*-OX and Sup52 plants show two bands (size indicated in bp) accounting for endogenous and transgenic *PLIP3* respectively while WT only shows one band. Two independent biological replicates are shown for each genotype.

purpose, we used whole genome sequencing data from *PLIP3*-OX homozygous lines as input for TC hunter and identified the insertion site for the transgene in *PLIP3*-OX on chromosome 5 (bp54108) in the 5' untranslated portion of gene AT5G01160 (Fig 2A). This locus is known to encode HAKAI, an E3 ubiquitin ligase involved in N6-adenosine methylation (m$^6$A) of mRNA in Arabidopsis and other organisms [53]. We confirmed this insertion point by PCR amplification of DNA from presumed homozygous *PLIP3*-OX plants with two primer pairs. The first pair contained two genome specific sequences designated left (LP1) and right (RP1), primers that hybridize to regions flanking the presumed insertion site (Fig 2B). We observed a single PCR product in WT samples but not in the *PLIP3*-OX transgenic line confirming the predicted insertion point and suggesting that the *PLIP3*-OX plants were homozygous for the T-DNA insertion (Fig 2C, upper panel). A second primer pair consisting of one T-DNA specific primer (LB1) and one of the genome specific primers (RP1) was used to confirm the presence of the T-DNA insertion in the *PLIP3*-OX plants (Fig 2C, lower panel). The expression of the *Hakai* gene was the same in WT and Sup52 (Fig 2D), but elevated in *PLIP3*-OX, suggesting that its expression might respond to the increased levels of JA in this line.

## A second-site mutation in Sup52 reverses the *PLIP3*-OX phenotype

Among the initial 90 putative mutants we focused on the Sup52 line which exhibited a clear partial suppression of the JA-induced phenotype (Fig 1A, S1 Fig). Quantitative phenotypic analyses showed that Sup52 plants had an increased rosette diameter and increased petiole length relative to *PLIP3*-OX, although not equal to WT (Fig 1A and B). Sup52 also ameliorated the anthocyanin accumulation in *PLIP3*-OX as reported by [8] and visible in vascular tissues. This was confirmed by phase partitioning during lipid extraction which showed a red pigment in the upper aqueous phase in *PLIP3*-OX and not in Sup52 (Fig 1C). Overexpression of *PLIP3* leads to a change in the lipid profile as previously described [8] and is diagnostic for the presence of a functional *PLIP3* transgene. Hence, to distinguish mutations in the transgene, which also could lead to phenotype reversion, from true second site suppressors, we routinely conducted lipid analyses. The acyl composition showed reduced 16:3 FAs in *PLIP3*-OX and Sup52 compared to WT (Fig 1D) indicating that the *PLIP3* gene was functional. Genotyping by PCR analysis of Sup52 plants using primers specific for the endogenous genomic *PLIP3* locus or the *PLIP3* cDNA in the transgenic *PLIP3*-OX line indicated the presence of the transgene (Fig 1E). Two amplified bands, one for endogenous genomic *PLIP3* and another for transgenic *PLIP3* locus were present in *PLIP3*-OX and Sup52 lines and only one band in WT corresponding to the endogenous genomic *PLIP3*. These results indicate that the *PLIP3* transgene is present in Sup52, its gene product remains active, and thus the reversion of the *PLIP3*-OX phenotype is due to a second-site mutation.

## A G776E substitution in the *LOX3* gene causes the reversion of the *PLIP3*-OX phenotype

To determine which second-site mutation in Sup52 causes *PLIP3*-OX phenotype supression, we backcrossed Sup52 to the homozygous parental *PLIP3*-OX line to generate an F1 line that after selfing yielded a F2 mapping population segregating for the second-site suppressor mutation. The examined F2 plants fell into three groups with 76 small plants, 113 intermediate and 74 large plants, a ratio indicative of a semidominant suppressor mutation (expected ratio 1:2:1, Chi$^2$ test p > 0.05). We extracted genomic DNA individually from all plants and pooled it in two groups: small and intermediate plants in one pool, and large plants in the second pool. The DNA concentration in each pool was normalized to adjust for equal DNA contribution of each plant in each pool. The two pools were submitted for whole genome, paired-end sequencing. The returned FASTA [54] formatted sequences were analyzed using the SIMPLE pipeline developed for causal mutation mapping [28]. SIMPLE generated a LOESS (locally estimated scatter plot smoothing) ratio plot showing chromosomal regions in which mutations co-segregate with suppressor phenotypes (S2 Fig). Of the candidates, the most probable causal mutation was a single G to A base substitution, at nucleotide position 2327 in the coding sequence of *LOX3*. This gene had been previously implicated in JA biosynthesis through its double mutant phenotype in combinations with mutations in other *LOX* genes [19,21,55]. It should be noted that *lox3* loss-of-function mutants by themselves have no reported phenotype to date, e.g.,

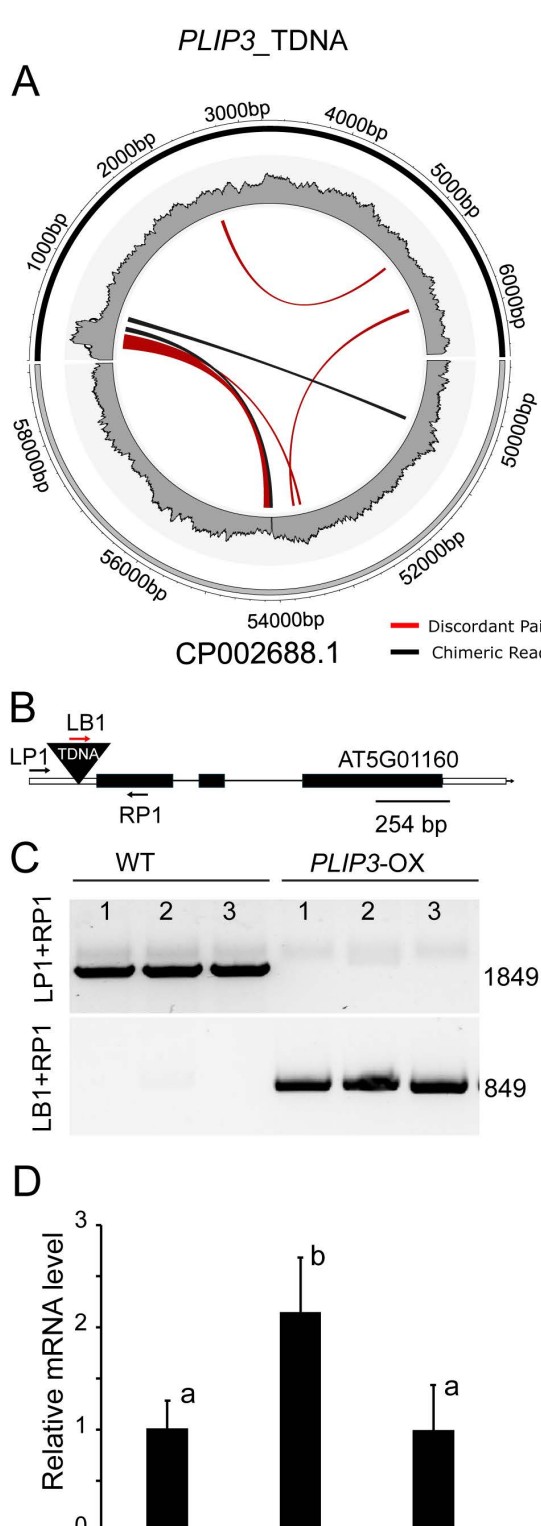

**A** *PLIP3*_TDNA

CP002688.1

■ Discordant Pair
■ Chimeric Read

**B**

LB1
LP1
TDNA
AT5G01160
RP1
254 bp

**C**

| | WT | | | *PLIP3*-OX | | |
|---|---|---|---|---|---|---|
| | 1 | 2 | 3 | 1 | 2 | 3 |

LP1+RP1 — 1849

LB1+RP1 — 849

**D**

**Fig 2. Determination of the *PLIP3*-OX T-DNA insertion site. (A)** The Insertion site of *PLIP3-OX* T-DNA was determined using TC-hunter. The bottom gray semi-circle represents the T-DNA insertion site on chromosome 5 (NCBI CP002688.1; bp position indicated) in the host while the upper black semi-circle depicts the sequence of the *PLIP3*-OX T-DNA. Multiple individual chimeric reads containing the insertion site are indicated with black lines,

discordant reads, i.e., those on either end of a DNA fragment crossing the insertion site generated during paired end sequencing, are indicated in red. Most reads fall at the left border of the T-DNA. A minority of reads mapping to different locations may be due to cloning artifacts during library preparation or sequencing inconsistencies. The inner circle gives an indication of the relative read number along the DNA sequence. **(B)** Schematic drawing of the *HAKAI* gene locus (AT5G01160) indicating the position predicted for the T-DNA insertion in the promoter region. Black boxes represent exons connected by lines which represent introns. Open boxes represent untranslated regions, which are only partially shown. Also indicated are the T-DNA as black triangle (not drawn to scale), and black arrows (not drawn to scale) are locations of primers LP1 and RP1, which are flanking the T-DNA insertion site on the left and right. The location of a T-DNA left border specific primer is shown as a red arrow (LB1, not drawn to scale). **(C)** Gel image showing PCR reactions using the T-DNA-specific primer and gene-specific primers as indicated in (B) confirming the insertion site identified by TC-hunter. **(D)** Relative expression of the *Hakai* gene determined by qPCR. Statistical analysis was performed in R using ANOVA followed by Tukey's multiple comparison test to compare relative expression in WT to that of *PLIP3*-OX and Sup52 plants. The different letters indicate a difference of the means with $p < 0.05$.

*lox3* mutants by themselves do not show male sterility [21]. The G to A towards the 3' end of the coding region (Fig 3A) leads to a change of glycine 776 to a glutamic acid residue in the C-terminal domain of the protein. We designated this new *lox3* mutant allele as *lox3–4*. Since *lox3–4* was generated through EMS mutagenesis, we could not yet rule out the possibility that additional independent background mutations could be involved in partially reversing the *PLIP3*-OX phenotype.

To determine whether the point mutation in *lox3–4* can cause suppression of the *PLIP3*-OX phenotypes as described in Fig 1, we crossed the homozygous *PLIP3*-OX line with homozygous *lox3–1* (SALK_119404; formerly designated *lox3A*) that carries a T-DNA insertion in the 4th intron (Fig 3A), and with homozygous *lox3–3* (SALK_062064, formerly designated *lox3D*) that harbors a T-DNA in the second intron [21] (S3A Fig). We adjusted the designation of the alleles to conform with the conventions for the Arabidopsis nomenclature [56]. The respective homozygous F2 plant phenotypes are shown in Fig 3B and Supplemental S3B Fig with an intermediate growth phenotype for the two *PLIP3*-OX *lox3* T-DNA alleles as was observed for the suppressor *PLIP3*-OX *lox3–4* line. The genotypes of the F2 plants were confirmed using two genome specific sequence primers, i.e., left (LP2) and right (RP2) primers for *lox3–1* (LP3 and RP3 for *lox3–3*) that hybridize to regions flanking the two T-DNAs in the two *lox3* alleles, respectively. We observed one PCR product in WT, *PLIP3*-OX, and *PLIP3*-OX *lox3–4* samples, but not in *PLIP3*-OX *lox3–1* and *lox3–1* lines (Fig 3C) or *PLIP3*-OX *lox3–3* and *lox3–3* lines (S3C Fig), respectively. Therefore, both *PLIP3*-OX lines contain the respective *lox3* T-DNA and these plants were homozygous for the T-DNA insertion. A second primer pair consisting of one T-DNA specific primer (LB2) and one of the genome specific primers (RP2 or RP3) was used to confirm the presence of the T-DNA insertion in the plants homozygous for *lox3–1* or *lox3–3* (Fig 3C and S3C Fig). The homozygosity of the *PLIP3* insertion locus in the *PLIP3*-OX lines was confirmed in these F2 plants using the LP1 and RP1 primers (see Fig 2; Fig 3D and S3B Fig) and the presence of genomic and cDNA copies of *PLIP3* was confirmed using primers P1 and P2, (Fig 3C and S3C Fig). To confirm that the *PLIP3* transgene retains the elevated PLIP3 lipase activity in these crosses, we profiled the leaf fatty acid composition of the different lines as a proxy for lipase activity. Indeed, overexpression of *PLIP3* caused a diagnostic reduction in 16:3 and concomitant increase in 16:0 fatty acids [8] (Fig 3D, S3D Fig), which is typical for homozygous *PLIP3*-OX lines indicating that the observed phenotype reversion in the suppressor lines was due to a second-site mutation and not a disruption of the *PLIP3* transgene. Taken together, the robust analysis discussed above of the selected *PLIP3*-OX lines carrying three different *lox3* mutant alleles, including the new *lox3–4* suppressor mutation corroborates that a loss of LOX3 in the *PLIP3*-OX background partially suppresses the growth phenotype. Comparing the phenotype of two T-DNA alleles with the *lox3–4* allele, we conclude that *lox3–4* is a loss of function allele, which will be discussed further below.

### *lox3–4* reverts *PLIP3*-OX phenotypes by reducing oxylipin biosynthesis

Given that *PLIP3*-OX phenotypes are caused by a constitutive JA response [8] and that LOXs catalyze the initial step of JA biosynthesis through oxygenation of 18:3 acyl groups, we reasoned that the partial suppression of *PLIP3*-OX phenotypes by introducing the *lox3–4* mutation is due to decreased metabolic flux through JA biosynthesis. To this end, we conducted a targeted metabolite analysis of leaf tissues of WT, *PLIP3*-OX, and *PLIP3*-OX *lox3–4* plants grown under standard physiological conditions. Indeed, JA, OPDA, methyl-JA (MeJA), and JA-Ile accumulated in *PLIP3*-OX more

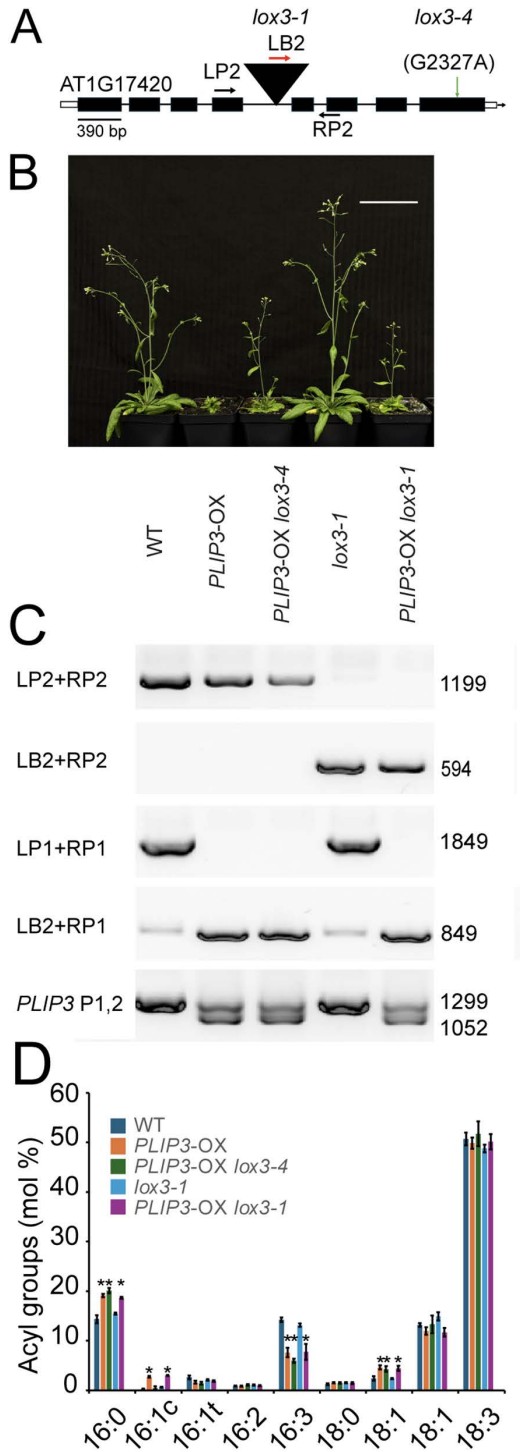

**Fig 3. A G2327A point mutation in *LOX3* is responsible for the *PLIP3*-OX suppressor phenotypes in the *lox3−4* mutant allele. (A)** Schematic drawing of the *LOX3* locus (AT1G17420) indicating the location of the G2327A point mutation in the *lox3−4* mutant allele. Black boxes represent exons, and black lines introns. The triangle indicates the T-DNA (not drawn to scale) in the *lox3−1* mutant allele. A left border primer in the T-DNA is indicated in red (LB2, not drawn to scale). Primers flanking the T-DNA insertion site in *lox3−1* (LP2, RP2) are indicated with black arrows (not drawn to scale). Open boxes indicate untranslated regions (only partly shown). **(B)** Phenotypes of six-week-old representative plants of WT, *PLIP3*-OX, *PLIP3*-OX *lox3−4* (aka Sup52), *lox3−1*, and the *PLIP3*-OX *lox3−1* mutant (from left to right). The plants were homozygous at all indicted loci. The scale bar represents 4 cm. **(C)**

Gel image of PCR genotyping results for the lines shown in (B) using primers as indicated in Fig 1B, the legend to Fig 2, and Fig 3A. The predicted fragment lengths are indicated (bp). **(D)**. The bar graph shows total fatty acyl composition (mol%) of the lines as indicated. n = 3, Student's t test was applied to compare plants with wild-type background, WT and *lox3−1*, to plants carrying the *PLIP3* overexpression construct, *PLIP3*-OX, *PLIP3*-OX *lox3−4,* and *PLIP3*-OX *lox3−1* (*P < 0.05); error bars show SD.

compared to WT, but not in *PLIP3*-OX *lox3–4* (Fig 4). The lower JA content of *PLIP3*-OX *lox3–4* was concomitant with reduced expression of JA-inducible genes compared to *PLIP3*-OX plants (S4 Fig). Taken together, our findings show that the G776E missense mutation in LOX3 suppresses *PLIP3*-OX phenotypes by partially inhibiting JA biosynthesis, leading to the decreased expression of JA responsive genes.

### The G776E substitution is predicted to impair LOX3 Function

Missense mutations change the amino acid sequence of proteins, which can have effects on the protein structure that lead to reduced function or even loss of function [57]. We hypothesized that the G776E substitution in *lox3–4* could significantly alter the structure of LOX3 and thus affect its capacity to catalyze the oxidation of polyunsaturated fatty acids to form hydroperoxides. LOX3 is a non-heme iron enzyme classified as 13-LOX because of its specificity for oxygenating C13 of 18:3 [18,19]. LOX3 has two domains. There is a N-terminal PLAT domain with a β barrel spanning AA residues 60–180 in the mature protein (AA residues 116–240 in the precursor protein). The C-terminal catalytic LOX α helix domain spans AA residues 181–863 in the mature protein (237–919 in the precursor protein). Thus, G776 is located within the catalytic domain of LOX3 (UniProt ID: Q9LNR3). Based on multiple sequence alignments using Clustal Omega [58] G776 of *At*LOX3 is conserved among different orthologs (Fig 5A), which suggests that this amino acid residue is likely important for functional or structural integrity of LOX3. Multiple sequence alignment of LOX3 with other Arabidopsis 13-LOX3s LOX2, LOX4 and LOX6 showed conservation of G776 and surrounding amino acids further highlighting the importance of this residue (S5B Fig). When using AlphaFold2 [59], it is apparent that in Arabidopsis LOX3 and its orthologs the Gly776 residue is within the vicinity of the catalytic non-heme iron (Fig 5B). Moreover, glycine and glutamate have profoundly different physical properties and thus substitution of one for the other can affect protein structure. With the proximity of G776 to the catalytic non heme iron and to the residues that position its ligands, substitution with the negatively charged Glu could alter the iron redox chemistry and thus impact alignment of substrate relative to the iron. Moreover, through high-resolution structural work, LOXs have been shown to rely on a rigid substrate tunnel and precise placement of residues around the non-heme iron and the pentadiene substrate [60]. Hence, the expected changes in geometry and charge resulting from the G776E substitution will likely negatively affect LOX activity.

### The *At*LOX3 G776E mutant protein remains stable but has an altered active site structure

To evaluate the effect of the mutation on the local dynamics of LOX3, including potential changes in binding activity, we developed structural models for the protein and its mutant version, and subsequently simulated them. We compared relevant quantities that can be tracked across the simulation set, such as the root mean square deviation (RMSD) to check for protein stability, as well as the root mean square fluctuation (RMSF) to evaluate especially dynamic parts of the protein.

The RMSDs for both the wildtype and G776E mutations are broadly similar across the simulations, with typical values between 2 and 4 Å and minimal rise during simulation after an initial equilibration period (S6B Fig). Such RMSD values are typical for folded proteins after thermalization, with many prior examples for RMSDs in this range in the literature, e.g., [61–63]. The modest RMSD values broadly indicate that the protein remains folded during simulation, with relatively small changes to the initial structure AlphaFold [59] predicted and AlphaFill [38] complemented during dynamics. This indicates that despite the lack of a crystal structure, the model proposed by these computational structure prediction tools are reasonable starting points to derive mechanistic insight as to the impact of the G776E mutation on LOX3 function.

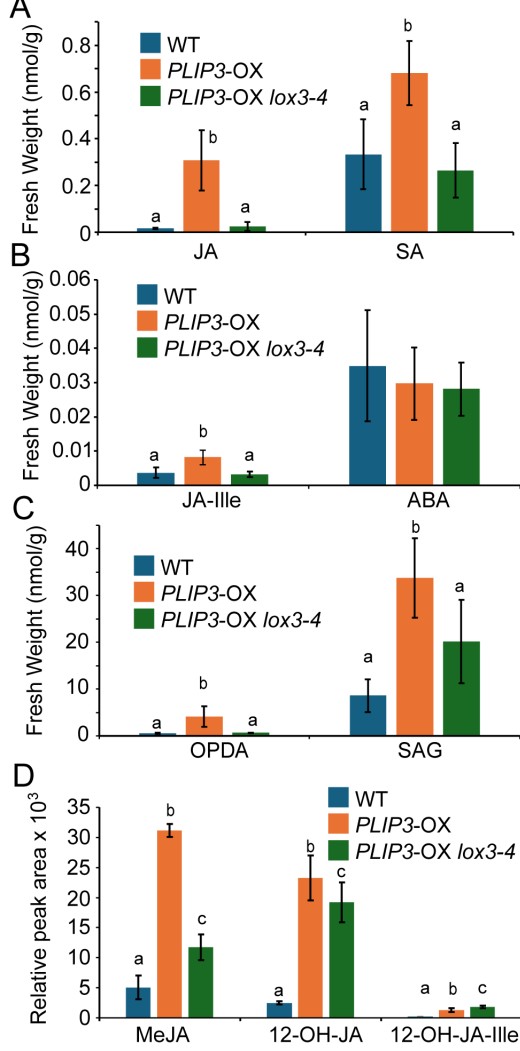

**Fig 4. Quantification of phytohormones and oxylipins by LC-MS/MS.** Phytohormones and oxylipins were extracted from 4-weeks-old plants. (A, B, C) Amounts quantified by LC-MS/MS for each genotype are shown. n = 5, error bars represent SD. Statistical analysis was performed in R using ANOVA followed by Tukey's multiple comparison test to compare the level in WT to that of *PLIP3*-OX and *lox3-4* plants. The different letters indicate a difference of the means with $p < 0.05$. In (D) the relative amounts were obtained for oxylipins for which there were no internal standards. n = 5 except for *PLIP3*-OX where n = 4; error bars show SD. Statistical analysis was performed in R using ANOVA followed by Tukey's multiple comparison test. The different letters indicate a difference of the means with $p < 0.05$. 12-OH-JA, 12-OH-jasmonic acid; 12-OH-JA-Ile 12-OH-JA, 12-OH-jasmonic acid-isoleucine; ABA, abscisic acid; JA, jasmonic acid; JA-Ile, jasmonic acid-isoleucine; OPDA, 12-oxo phytodienoic acid; SA, salicylic acid; SAG, salicylic acid glycoside.

While the overall protein structure remained properly folded regardless of the mutation (SFig. 6B), the simulations offered hints regarding the mechanistic reasons for the observed change in function in vivo. When we quantified the RMSF (Fig 6A), most residues of LOX3 fluctuated similarly across both WT and G776E mutant simulations. Zooming into the differences (Fig 6B), there were two regions with sharply higher fluctuations in the mutant. The first region immediately followed the mutation site in the sequence, and likely was directly driven by the glutamate taking up more space than the glycine it replaces and distorting the local protein structure. The second region with increased fluctuation was near the C-terminus and was spatially close to the mutation site (Fig 6C, S6A Fig). Other regions further from the mutation site also exhibit smaller peaks and valleys along the RMSF profile (Fig 6B).

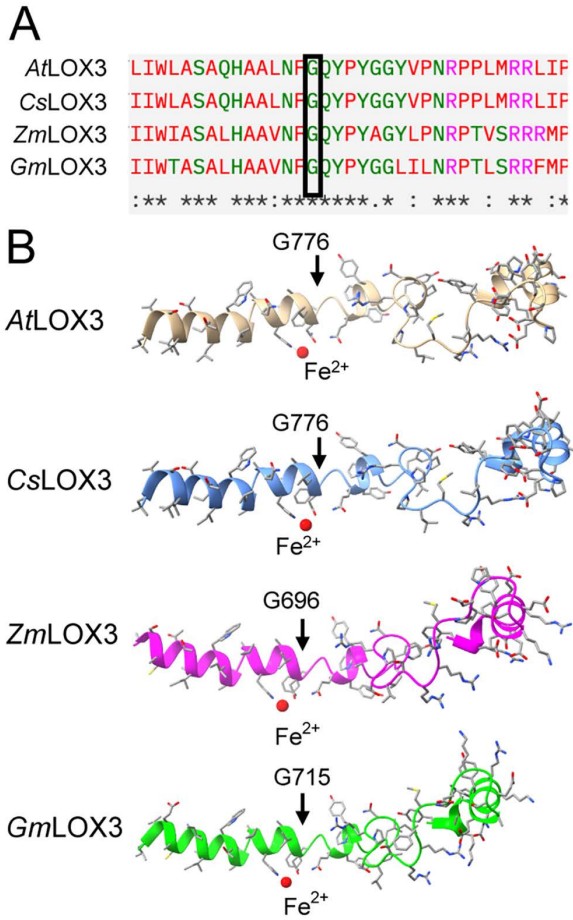

## A

```
AtLOX3  LIWLASAQHAALNFGQYPYGGYVPNRPPLMRRLIF
CsLOX3  IIWLASAQHAALNFGQYPYGGYVPNRPPLMRRLIF
ZmLOX3  IIWIASALHAAVNFGQYPYAGYLPNRPTVSRRRMF
GmLOX3  IIWTASALHAAVNFGQYPYGGLILNRPTLSRRFMF
        :** *** ***:*******.* :  *** :  ** :*
```

## B

**AtLOX3** G776 Fe²⁺

**CsLOX3** G776 Fe²⁺

**ZmLOX3** G696 Fe²⁺

**GmLOX3** G715 Fe²⁺

**Fig 5. Sequence alignment of Arabidopsis LOX3 with other orthologs. (A)** Alignment of a portion of LOX3 protein sequences from different plant species. *At: Arabidopsis thaliana*, (OAP14869.1); *Cs: Camelina sativa* (XP_010476891.1); *Zm: Zea mays* (NP_001105515.1); *Gm: Glycine max* (NP_001235383.2). The *lox3-4* mutation affects a glycine residue that is conserved in the catalytic domain (black box). **(B)** AlphaFold 2 generated models for the portion of the polypeptide chain showing the position of the mutated glycine in *lox3-4* relative to the non heme iron. This residue is conserved in all species shown.

Focusing on the C-terminal region, a reasonable mechanistic hypothesis is that additional fluctuations in this region disrupt iron binding to LOX3, impairing its function. The C-terminal carboxyl group is predicted to coordinate the iron, and this iron is likely responsible for the lipoxygenase activity. If the residues immediately upstream of the C-terminus fluctuate more than they otherwise would be due to the G776E mutation, it stands to reason that the C-terminus may not stably coordinate the iron. The change in iron coordination without C-terminal interactions may reduce binding or the electronic environment around the iron may no longer be able to facilitate catalysis. From a molecular simulation perspective, this is difficult to prove, as the interactions between iron and the rest of the protein are not well captured by typical classical force fields, and quantum methods cannot readily reach timescales where the protein conformation could substantially change. Indeed, the mutated residue G776E is on the same helix (but on the opposite side) as H770, a residue also directly involved in coordinating the iron. Our simulations assumed that iron coordinating residues are covalently bonded to the iron and thus showed normal RMSF. However, a newly synthesized LOX3 polypeptide with the G776E mutation may not bind and retain an iron ligand, as the C-terminal tail will fluctuate more and hinder tight iron coordination.

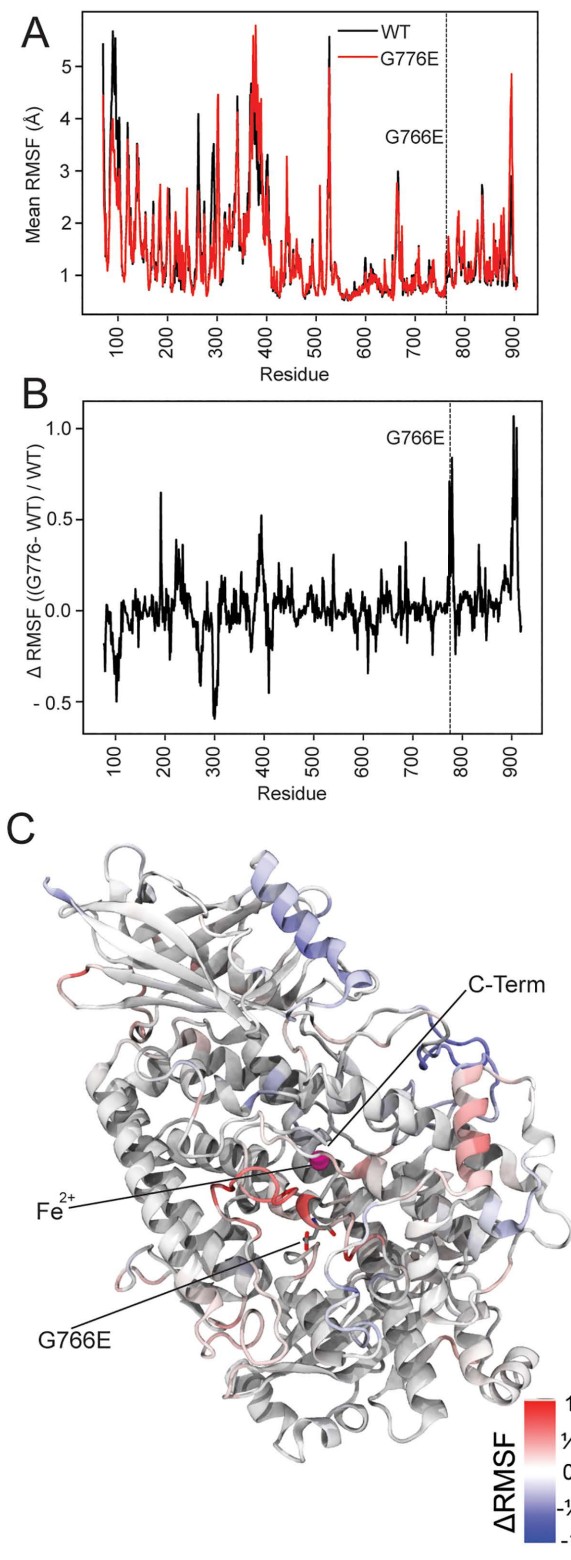

**Fig 6. Comparison of the structural integrity of Arabidopsis WT LOX3 and the mutant protein in *lox3-4*.** The root mean-square fluctuation (RMSF) values for both the WT and mutant *lox3-4* structures (A) are broadly similar across most of the protein. However, evaluating the difference between the RMSF values of both structures **(B)**, there are regions with substantially elevated RMSF near the C-terminus. Taking the values from panel (B) to color the protein structure in panel **(C)**, these regions are spatially near the site of the G776E mutation. The mutation site, the iron (red sphere), and the

C-terminal α-carbons is indicated. The protein is drawn with higher variability locations in the mutant version colored red, while protein regions with lower variability in the mutant protein are colored blue.

An alternative mechanism, that the G776E mutation alters substrate binding, is less likely from the simulations. Reactive metal centers often confine their substrates to the active site or to tunnels that lead to the active site [64]. Residue 776 is on the wrong side of the helix to be engaged with a substrate that interacts directly with the iron. For instance, AlphaFill places nitrocatechol inside a pocket within LOX3 that is directly adjacent to the active site whereas residue G776E faces solution.

Finally, we considered the possibility that the expression of the lox3–4 allele is altered in PLIP3-OX lox3–4 resulting in lower LOX3 activity. However, the expression of the lox3–4 alle does not seem to be different in the PLIP3-OX background compared to the wild-type allele in the WT or the PLIP3-OX line as shown in S7 Fig. This suggests that the reduced activity of the enzyme encoded by lox3–4 as suggested by the molecular dynamics analysis above is the cause of the suppressor phenotype.

## Conclusions

We identified a new allele, lox3–4, causing the loss of a 13S-lipoxygenase involved in JA biosynthesis in Arabidopsis. Previously, lox3 had not shown phenotypes except in combination with other lox mutations. It seems likely that PLIP3-OX is unique in providing a steady-state level of JA that is responsive to slight variations in enzyme activities associated with its synthesis. It is also possible that LOX3 has direct access to the acyl groups released by PLIP3 while the other LOX enzymes have not, possibly due to compartmentalization of the involved enzymes within the chloroplast, for example inside the thylakoids versus the stroma. We have noticed that the mutant phenotypes are attenuated depending on the zygosity at the transgene locus as well as at the lox mutant locus. Thus, the contribution of a single lipoxygenase can manifest itself in this system. Therefore, a strategy has emerged to investigate the contribution of other isoforms of enzymes involved in oxylipin biosynthesis in this PLIP3-OX background.

## Supporting information

**S1 Fig. Phenotypes of 6-week-old plants grown under standard conditions.** Shown are representative plants of wild type (WT), PLIP3-OX homozygous overexpression line (PLIP3-OX) and the #52 suppressor mutant line in the PLIP3-OX homozygous background (Sup52). The scale bar indicates 4 cm. (B) Leaf diameter of 6-week-old plants. N = 20, error bars indicate SD. Statistical analysis was performed in R using ANOVA followed by Tukey's multiple comparison test to compare the leaf diameter in WT to that of PLIP3-OX and Sup52 plants. The different letters indicate a difference of the means with $p < 0.05$. (C) Anthocyanin levels in 4-week-old plants. N = 20, error bars indicate SD. Statistical analysis was performed in R using ANOVA followed by Tukey's multiple comparison test to compare the leaf diameter in WT to that of PLIP3-OX and Sup52 plants. The different letters indicate a difference of the means with $p < 0.05$. Results were normalized to WT = 1.
(TIF)

**S2 Fig. Read out of the SIMPLE pipeline.** Shown is the chromosome location on the X-axis and the LOESS (locally estimated scatter plot smoothing) ratio variable as defined in [28], indicting the probability for the causal mutation for the selected phenotype. The opposition of the prime candidate LOX3 on chromosome 1 is indicated.
(TIF)

**S3 Fig. A G2327A point mutation in LOX3 is responsible for the PLIP3-OX suppressor phenotypes in the lox3−4 mutant allele.** (A) Schematic drawing of the LOX3 locus (AT1G17420) indicating the location of the G2327A point

mutation in the *lox3−4* mutant allele. Black boxes represent exons, and black lines introns. The triangle indicates the T-DNA (not drawn to scale) in the *lox3−3* mutant allele. A left border primer in the T-DNA is indicated in red (LB2, not drawn to scale). Primers flanking the T-DNA insertion site in *lox3−3* (LP3, RP3) are indicated with black arrows (not drawn to scale). Open boxes indicate untranslated regions (only partly shown). (B) Phenotypes of six-week-old representative plants of WT, *PLIP3*-OX, *PLIP3*-OX *lox3−4* (aka Sup52), *lox3−3*, and the *PLIP3*-OX; *lox3−3* mutant (from left to right). The plants were homozygous at all indicted loci. The scale bar represents 4 cm. (C) Gel image of PCR genotyping results for the lines shown in (B) using primers as indicated in Fig 1B, the legend to Fig 2, and S3A Fig. The predicted fragment lengths are indicated (bp). (D) The bar graph shows total fatty acyl composition (mol%) of the lines as indicated. n = 3, Student's t test was applied to compare plants with wild-type background, WT and *lox3−1*, to plants carrying the *PLIP3* overexpression construct, *PLIP3*-OX, *PLIP3*-OX *lox3−4,* and *PLIP3*-OX *lox3−1* (*P < 0.05); error bars show SD.
(TIF)

**S4 Fig. Expression of JA responsive genes is reduced in *PLIP3*-OX *lox3–4*.** Real time quantitative PCR (RT-qPCR) was used to quantify the expression of the three JA-responsive genes *VSP1* (At5g24780), *PDF1.2* (At5g44420), and *LOX2* (AT3G45140) in the three lines as indicated. Statistical analysis (n = 3) was performed in R using ANOVA followed by Tukey's multiple comparison test to compare relative expression of the genes as indicated in WT, *PLIP3*-OX, and *PLIP3*-OX *lox3–4* plants The different letters indicate a difference of the means with p < 0.05; error bars show SD.
(TIF)

**S5 Fig. Sequence conservation of Arabidopsis LOX paralogs.** Alignment of a portion of the *At*LOX3 protein with the corresponding sequences for *At*LOX2 (AT3G45140), *At*LOX4 (AT1G72520) and *At*LOX6 (AT1G67560). The *lox3–4* mutation affects a glycine residue that is conserved in the catalytic domain (black box).
(TIF)

**S6 Fig. The G776E mutation in *lox3–4* does not lead to an unfolding of the *AT*LOX3 protein but affects the active site structure.** (A) Active site detail of the LOX3 structure shown in Figure 6C. The root mean-square fluctuation (RMSF) difference values for the WT and mutant *lox3–4* structures show increased RMSF values for the G776E mutant protein near the C-terminus, which is close to the active site. The structure is drawn with higher variability locations in the mutant version which are colored red, while protein regions with lower variability in the mutant protein are colored blue. In addition to the labels in Figure 6C, the two active site histidines, H770 and H578, are pointed out in blue. They coordinate the iron (red sphere). (B) Molecular dynamics comparison of the structural integrity of the *AT*LOX3 WT and G776E mutant protein indicates that the protein structure remains intact. The root mean-square deviations for the two structures are shown for four simulations, each over 1000 ns.
(TIF)

**S7 Fig. Relative expression of the *LOX3* gene determined by qPCR.** Statistical analysis was performed in R using ANOVA followed by Tukey's multiple comparison test to compare relative expression in WT to that of *PLIP3*-OX, *PLIP3*-OX *lox3−4*, and *lox3−1* plants. The different letters indicate a difference of the means with p < 0.05.
(TIF)

**S1 Table. Primers used in the study.**
(DOCX)

## Acknowledgments

We would like to express our gratitude to the students who participated in the mutant screening during a course-based undergraduate research experience (CURE) for their enthusiasm, engagement, and dedication. In particular, we thank the following undergraduate students that participated in this project in the Benning laboratory: Michael Beecher, Jordyn

Flemming, and Halle Purcell. We thank Dr. Tony Schillmiller and James O'Keefe from the MSU Research Technology Support Facility for his help with targeted metabolite analysis. We also thank Dr. Nicholas Panchy, MSU Institute for Cyber-Enabled Research, Bioinformatics Core (RRID:SCR_026706), for his help with implementing TC-Hunter. We also thank Joanne Thomson for help with setting up the qPCR experiment.

## Author contributions

**Conceptualization:** Yosia Mugume, Ron Cook, Jinjie Liu, John E Froehlich, Josh V. Vermaas, Christoph Benning.

**Data curation:** Yosia Mugume.

**Formal analysis:** Yosia Mugume, Ron Cook, Breana Hagerty, Zachary B. Alvord, Linda Danhof, Josh V. Vermaas, Christoph Benning.

**Funding acquisition:** Christoph Benning.

**Investigation:** Yosia Mugume, Ron Cook, Breana Hagerty, Jinjie Liu, Linda Danhof, John E Froehlich.

**Methodology:** Linda Danhof.

**Project administration:** Christoph Benning.

**Supervision:** Josh V. Vermaas, Christoph Benning.

**Visualization:** Zachary B. Alvord, Josh V. Vermaas.

**Writing – original draft:** Yosia Mugume.

**Writing – review & editing:** Ron Cook, Jinjie Liu, John E Froehlich, Josh V. Vermaas, Christoph Benning.

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
