## [Decision Letter · Decision Letter 0]

10 Mar 2026

PONE-D-26-02367A Lipoxygenase 3 mutation reverses growth phenotypes in an Arabidopsis Plastid Lipase 3 overexpression linePLOS One

Dear Dr. Benning,

Thank you for submitting your manuscript to PLOS ONE. After careful consideration, we feel that it has merit but does not fully meet PLOS ONE’s publication criteria as it currently stands. Therefore, we invite you to submit a revised version of the manuscript that addresses the points raised during the review process.

A letter that responds to each point raised by the academic editor and reviewer(s). You should upload this letter as a separate file labeled ’Response to Reviewers’.A marked-up copy of your manuscript that highlights changes made to the original version. You should upload this as a separate file labeled ’Revised Manuscript with Track Changes’.An unmarked version of your revised paper without tracked changes. You should upload this as a separate file labeled ’Manuscript’.

We look forward to receiving your revised manuscript.

Kind regards,

Hiroshi Ezura

Academic Editor

PLOS One

Journal Requirements:

1. Please ensure that your manuscript meets PLOS ONE’s style requirements, including those for file naming. The PLOS ONE style templates can be found at

“This work was primarily supported by the National Science Foundation (MCB-2203474) and in part by the Office of Basic Energy Sciences of the United States Department of Energy (Grant DE-FG02-91ER20021) and by MSU AgBioResearch (MICL02632). Z.B.A. was supported by the High School Honors Science, Math and Engineering Program at Michigan State University. This work was supported in part through computational resources and services provided by the Institute for Cyber-Enabled Research at Michigan State University.”

“This work was primarily supported by the National Science Foundation (MCB-2203474) and in part by the Office of Basic Energy Sciences of the United States Department of Energy (Grant DE-FG02-91ER20021) and by MSU AgBioResearch (MICL02632). Z.B.A. was supported by the High School Honors Science, Math and Engineering Program at Michigan State University. This work was supported in part through computational resources and services provided by the Institute for Cyber-Enabled Research at Michigan State University.”

Reviewers’ comments:

Reviewer’s Responses to Questions

**Comments to the Author**

1. Is the manuscript technically sound, and do the data support the conclusions?

Reviewer #1: Yes

Reviewer #2: Yes

2. Has the statistical analysis been performed appropriately and rigorously? 

Reviewer #1: Yes

Reviewer #2: Yes

3. Have the authors made all data underlying the findings in their manuscript fully available?

Reviewer #1: Yes

Reviewer #2: Yes

4. Is the manuscript presented in an intelligible fashion and written in standard English?

Reviewer #1: Yes

Reviewer #2: Yes

5. Review Comments to the Author

Reviewer #1: The manuscript by Mugume et al. reports a mutation in LOX3 that attenuates the effects of PLIP3 overexpression on the plant growth and on jasmonic acid production in Arabidopsis thaliana. The manuscript is clearly and concisely written based on the results, and no major concerns were identified. I have only one comment as follows.

In lines 93–99, the authors state that, to query the entire pathway from the release of fatty acid precursors through the biosynthesis of JA and other oxylipins to their subsequent perception, transduction, and modification by other signaling pathways, they performed a suppressor screen in the PLIP3-OX transgenic line. However, I could not find a discussion of this question in the manuscript, particularly in light of the enzymatic characteristics of LOX3 and PLIP3 and their potential interactions.

Wang et al. (2018) suggest that PLIP3 preferentially targets phosphatidylglycerol in vivo, but its overexpression substantially decreases the 16:3 fatty acid content (Fig. 1D), which is specific to MGDG. This observation may indicate that overexpressed PLIP3 releases 18:3 from the sn-1 position of MGDG, which could then be utilized by LOX3 and subsequently for OPDA production. Nonetheless, in chloroplasts, the major LOX enzyme LOX2, and presumably LOX4 and LOX6 as well, are also capable of 13S-lipoxygenation. It is therefore unclear why the loss of LOX3 strongly suppresses JA production in the PLIP3-OX background, given that LOX2 and other LOXs would be expected to metabolize the 18:3 fatty acids released by PLIP3. Are the fatty acids excised by PLIP3 not suitable substrates for these LOX enzymes? Please include a discussion on the genetic and possible biochemical interactions between LOX3 and PLIP3 in relation to the biosynthesis of JA and other oxylipins.

Reviewer #2: This manuscript reports a suppressor screen to dissect the dwarf and abnormal growth phenotypes caused by PLIP3 overexpression (PLIP3-OX) and identifies a suppressor as a new LOX3 allele (lox3-4; G776E). The overall story is easy to follow, and the genetic case is strengthened by the fact that independent lox3 T-DNA alleles (lox3-1 and lox3-3) also partially suppress PLIP3-OX phenotypes. The hormone/oxylipin trends are generally consistent with the proposed interpretation.

A few issues should be addressed to strengthen the conclusions. First, the PLIP3-OX insertion maps near AT5G01160 (HAKAI). While the authors note this, the manuscript still largely assumes the phenotype reflects PLIP3 overexpression alone. It would help to show that HAKAI expression is not perturbed in PLIP3-OX (e.g., RT-qPCR), or, if available, to include data from an independent PLIP3 overexpression line. At a minimum, the limitations around insertion-site effects should be stated more explicitly.

Second, the discussion of the lox3-4 mutation would benefit from clearer weighting of evidence. The structural prediction/MD analysis is interesting and offers a plausible explanation for how G776E could compromise LOX3 function, but without direct biochemical support (LOX activity measurements, product profiling, or LOX3 protein abundance/stability), it remains speculative. In its current form, I would place the MD/structure analysis in the Supplementary Information and keep the main text to a brief, cautious statement (e.g., “consistent with reduced LOX3 function”), unless functional validation is added. Related to this, LOX3 transcript (and ideally protein) levels would help distinguish expression/stability effects from catalytic impairment, and a simple 13-LOX activity assay would provide a direct test. If that is not feasible, a more explicit quantitative comparison of JA/OPDA and related oxylipins across lox3-4 and established lox3 null/strong alleles would still strengthen the argument.

Third, given the EMS origin of lox3-4, it would be helpful to state more clearly how background mutations were handled (e.g., number of backcross generations and whether key phenotypes/hormone profiles were reproduced after backcrossing). The replication with T-DNA lox3 alleles supports causality, but a short clarification here would improve transparency.

Finally, the statistics and reporting could be tightened. Several panels appear to use multiple pairwise t-tests; ANOVA with an appropriate post hoc test (or an explicit multiple-comparison correction) would be more appropriate. Please also report sample sizes unambiguously (avoid “n = 4 or 5” without specifying which datasets correspond to each n).

Minor points: The screen mentions multiple phenotypic features, but the figures emphasize petiole length; adding rosette diameter/biomass (or summarizing key metrics) would make “partial recovery” more convincing. Growth conditions (age, photoperiod, light intensity) should be reported consistently across figures. The anthocyanin-related observation based on the reddish phase in lipid extraction would be more convincing with a simple quantification. Please also correct minor typos (e.g., “lipoxgenase” → “lipoxygenase”) and ensure consistent terminology throughout.

6. PLOS authors have the option to publish the peer review history of their article (what does this mean?). If published, this will include your full peer review and any attached files.

Reviewer #1: No

Reviewer #2: No

---

## [Editor Report · Decision Letter 1]

18 May 2026

A Lipoxygenase 3 mutation reverses growth phenotypes in an Arabidopsis Plastid Lipase 3 overexpression line

PONE-D-26-02367R1

Dear Dr. Benning,

We’re pleased to inform you that your manuscript has been judged scientifically suitable for publication and will be formally accepted for publication once it meets all outstanding technical requirements.

An invoice will be generated when your article is formally accepted. Please note, if your institution has a publishing partnership with PLOS and your article meets the relevant criteria, all or part of your publication costs will be covered. Please make sure your user information is up-to-date by logging into Editorial Manager at Editorial Manager® and clicking the ‘Update My Information’ link at the top of the page. For questions related to billing, please contact billing support.

Kind regards,

Hiroshi Ezura

Academic Editor

PLOS One

Additional Editor Comments (optional):

Reviewers’ comments:

---

## [Editor Report · Acceptance letter]

PONE-D-26-02367R1

PLOS One

Dear Dr. Benning,

I’m pleased to inform you that your manuscript has been deemed suitable for publication in PLOS One. Congratulations! Your manuscript is now being handed over to our production team.

Lastly, if your institution or institutions have a press office, please let them know about your upcoming paper now to help maximize its impact. If they’ll be preparing press materials, please inform our press team within the next 48 hours. Your manuscript will remain under strict press embargo until 2 pm Eastern Time on the date of publication. For more information, please contact onepress@plos.org.

Kind regards,

on behalf of

Prof. Hiroshi Ezura

Academic Editor

PLOS One